# Potential Role of Carbon Nanomaterials in the Treatment of Malignant Brain Gliomas

**DOI:** 10.3390/cancers15092575

**Published:** 2023-04-30

**Authors:** Maria Caffo, Antonello Curcio, Kumar Rajiv, Gerardo Caruso, Mario Venza, Antonino Germanò

**Affiliations:** 1Department of Biomedical and Dental Sciences and Morphofunctional Imaging, Neurosurgical Clinic, University of Messina, 98125 Messina, Italycrcnnl91s28e977t@studenti.unime.it (A.C.);; 2NIET, National Institute of Medical Science, New Delhi 110007, India; 3University of Delhi, New Delhi 110007, India

**Keywords:** blood–brain barrier, brain drug delivery, carbon nanomaterials, cerebral gliomas, glioblastoma, nanoparticles

## Abstract

**Simple Summary:**

Nanomaterials are one of the most promising discoveries of this millennium. Thanks to the widest range of applications, this field has spread to all scientific disciplines. As well, interest has significantly increased in the medical sector. Although there are many different families of nanoparticles, carbon-based nanoparticles have only recently come to light. An infinite number of nanoparticles of various shapes and sizes can be produced by taking advantage of the chemical bonding properties of carbon, which also allows for the modification of their chemical, thermal, and physical properties. This review examines the biomolecular aspects of the theoretical and practical challenges involved in creating nanoparticles with biological activity, identifying the benefits and drawbacks of each approach, and summarizing the most recent research on carbon-based nanoparticles conceptualized and developed to date. Although it is a very promising area of study, more pharmacokinetic and toxicological research is still required.

**Abstract:**

Malignant gliomas are the most common primary brain tumors in adults up to an extent of 78% of all primary malignant brain tumors. However, total surgical resection is almost unachievable due to the considerable infiltrative ability of glial cells. The efficacy of current multimodal therapeutic strategies is, furthermore, limited by the lack of specific therapies against malignant cells, and, therefore, the prognosis of these in patients is still very unfavorable. The limitations of conventional therapies, which may result from inefficient delivery of the therapeutic or contrast agent to brain tumors, are major reasons for this unsolved clinical problem. The major problem in brain drug delivery is the presence of the blood–brain barrier, which limits the delivery of many chemotherapeutic agents. Nanoparticles, thanks to their chemical configuration, are able to go through the blood–brain barrier carrying drugs or genes targeted against gliomas. Carbon nanomaterials show distinct properties including electronic properties, a penetrating capability on the cell membrane, high drug-loading and pH-dependent therapeutic unloading capacities, thermal properties, a large surface area, and easy modification with molecules, which render them as suitable candidates for deliver drugs. In this review, we will focus on the potential effectiveness of the use of carbon nanomaterials in the treatment of malignant gliomas and discuss the current progress of in vitro and in vivo researches of carbon nanomaterials-based drug delivery to brain.

## 1. Introduction

Cerebral gliomas are the most frequent, intrinsic, primary tumors of the central nervous system (CNS). Their incidence is about 6 cases per 100,000 people per year [1]. Among them, glioblastoma (GB) is the most frequent and most malignant histological type, the incidence of which is approximately 57% of all gliomas and 48% of all primary malignant CNS tumors [2]. It predominantly affects adults with a maximum incidence between 50 years and 70 years. Gliomas observed after the age of sixty account for 90% of GB. The grading system, recently updated, proposed by WHO is the most accepted and widespread [3]. The new WHO classification combines, in addition to data relating to tumor histology and grading, also molecular data, thus obtaining a system for evaluating brain tumors, much more precise and intrinsically linked to the biomolecular characteristics of the specific cancer. GB has rapid growth that is expansive and infiltrates the surrounding nervous parenchyma. Rapid evolution with the appearance of focal syndrome associated with signs of intracranial hypertension is a common clinical sign.

Surgery still remains the first step of treatment in gliomas, as it is also necessary to obtain a definitive histological examination of the lesion. Usually, the surgery must aim at the removal of the tumor as radically as possible (Figure 1). Alternatively, only a biopsy can be performed. However, rare cases of gliomas of the diencephalon, midbrain, and deep hindbrain and those with extensive extension into the corpus callosum are not amenable to surgical treatment. STUPP protocol is based on the administration of temozolomide (TMZ) followed by radiotherapy (RT). TMZ is an oral alkylating agent that is a prodrug that activates itself, without enzymatic catalysis in the physiological pH of cells, into the active metabolite monomethyl triazenoimidazole carboxamide (MTIC). The toxic effects of MTIC are associated with alkylation of DNA, especially at the O6 and N7 positions of the nitrogenous base guanine. RT consists of a total of 60 Gy fractionated, targeted, doses of 2 Gy once daily for five days a week for six weeks. The chemosensitizing protocol contemplates the administration of 75 mg per m^2^ per day, every day until the end of the radiotherapy. Subsequently, one month later, six/twelve cycles of chemotherapy began. Each cycle is defined as 5 days of TMZ every 28 days at a maximum dose of 150 mg per m^2^ per day for the first cycle and 200 mg per m^2^ per day for the next 5 cycles. The STUPP protocol increased median survival to 14.6 months versus 12.1 with radiotherapy alone. The five-year survival rate increased to 9.8% versus 1.9% without STUPP protocol [4,5]. Another potentially useful drug in recurrent GBM is bevacizumab, a monoclonal antibody against VEGF. Irinotecan and bevacizumab demonstrated notable antitumor activity in patients with GBM, already surgically treated, in first or second relapse [6]. Procarbazine, lomustine, and vincristine (PVC) are indicated as second line in patients with poor response to the STUPP protocol. However, the efficacy of current anti-cancer strategies in gliomas is limited by the blood–brain barrier (BBB) that hinders the delivery of many chemotherapeutic agents and macromolecules. Tumoral invasion is a multifactorial process, characterized by interactions between extracellular matrix protein and adjacent cells, as well as accompanying biochemical processes supportive of active cells movement [7]. Recent advances in gliomas molecular pathology and biology have evidence of the various genes involved in cell growth, apoptosis, and angiogenesis. The modulation of gene expression at more levels, such as DNA, mRNA, proteins, and transduction signal pathways, may be the most effective modality to down-regulate or silence some specific genes functions.

Nanotechnology, which is widely used in many industrial trades, can be a valuable aid in the development of new glioma treatments. Because of their size, nanoparticles (NPs) can cross the BBB and, by acting as carriers, can deliver even more therapeutic compounds capable of interacting with multiple targets. It is possible to use nanotechnology to deliver the drug to the targeted tissue across the BBB, release the drug at a controlled rate, and avoid multidrug resistance. NPs can be designed to transport therapeutic drugs and imaging agents that are loaded onto or within the nanocarriers via chemical conjugation or encapsulation.

Carbon nanomaterials (CNs), which have been studied for some time, possess peculiar characteristics such as long stability and the ability to form stable bonds with various functional groups so as to make them suitable for numerous applications in both the industrial and biomedical fields [8]. The high biocompatibility makes them particularly functional in medical and pharmacological technologies. CNs also possess antibacterial activity [9] and can be structured into pharmacological compounds with potential use both in new anticancer therapeutic protocols and in brain drug delivery systems [10].

Aim of this study was to show the potential and most innovative applications of CNs in the treatment of brain tumors.

## 2. Nanotechnology

Nanotechnology is regarded as a developing field with potential applications in cancer research and treatment. The manipulation of matter at the molecular and atomic levels (i.e., on a dimensional scale smaller than the micrometer, i.e., between 1 and 100 nanometers) is the domain of nanotechnologies. In comparison to conventional treatments, NP systems in cancer therapies provide better therapeutic and diagnostic agent penetration and lower risk [11]. Many mechanisms for brain-targeted delivery can be engineered into NPs, including receptor-mediated transcytosis, carrier-mediated transcytosis, and adsorptive-mediated transcytosis. These systems can also reduce toxicity to peripheral organs and improve biodegradability. The goal of nanotechnology is to create and characterize ultra-small particles. NPs are structures with a diameter of 10–200 nm that have nearly limitless design and application possibilities in biologic systems and are used for diagnostic and therapeutic purposes. NPs can penetrate cell membranes and collaborate with biomolecules due to their extremely small dimensions, and their physical properties make them excellent imaging agents and semiconductors. The goal of using NPs for cancer treatment is to deliver the right drugs to the right patient at the right time in the right concentration [12]. This ideal concept is difficult to achieve due to the disparities in drug adhesion, distribution, metabolism, and excretion [13]. Many factors make nanomedicine superior to conventional medicine for cancer treatment: because of the increased permeability of malformed tumor vascular walls with leaky cell-to-cell junctions and dysfunctional lymphatics in tumorous tissues, their dimensions allow them to be passively accumulated in cancer cells; the expression on their surface of various targeting ligands allows the link with specific targets on tumor cells or in tumor microenvironment (TME), enhancing their accumulation [14]. Because NPs are not physically recognized as substrates, they can be used to bypass tumor escape mechanisms as drug efflux pumps [15]. In vivo and ex vivo studies show that NPs are more useful for detecting and killing cancer cells due to the delivery and release of bioactive molecules under desired temperature, pH, or enzymatic catalysis conditions [16,17]. Furthermore, encapsulation protects bioactive molecules from degradation, increasing their solubility in biological fluids. The transport of NPs through blood circulation to tumor regions via blood vessels, the crossing of vasculature walls to reach surrounding tumor tissues, the introduction in the interstitial space to target cells, and cellular uptake via endocytosis and intracellular delivery are all part of the in vivo NP delivery process [18]. Phagocytosis, clathrin-mediated endocytosis, caveolin-mediated endocytosis, clathrin/caveolae-independent endocytosis, and micropinocytosis are the five major mechanisms involved in the endocytosis of NPs by target cells. The process of releasing a compound at a specific rate and location is known as drug delivery. Novel drugs need effective delivery technologies that reduce side effects and improve patient compliance. Conventional anticancer agents are cytotoxic due to their low molecular weights and high pharmacokinetic volumes of distribution. High concentrations yield effective doses, but when administered alone, these drugs lack specificity and cause significant damage to non-cancerous tissues. Furthermore, the majority of chemotherapeutic agents are poorly soluble and are mixed with toxic solvents [19]. NP-based drug delivery systems improve the penetration of therapeutic and diagnostic agents into the desired site, allowing for efficacy with lower doses and systemic drug concentration with minimal risks. NP-based drug delivery has the potential to improve drug bioavailability, improve drug molecule timing, and enable precision drug targeting without compromising the structural and functional integrity of the BBB [20]. Size-dependent passive targeting or active targeting can be used to deliver NPs to specific sites. Passive targeting entails chemically modifying the NPs to increase permeability or stability. Insertion of ethylene oxide polymers, also known as poly-(ethylene glycol) (PEG), is the most common surface modification. PEG can increase the half-life of nanocarrier drug delivery systems by decreasing macrophage uptake due to steric repulsion effects and inhibiting plasma-protein adsorption [21]. PEGylation has been used successfully in a wide range of drug delivery systems, including lipid, polymeric, and inorganic NPs. Active targeting is typically accomplished through the incorporation of a receptor-specific ligand that promotes the targeting of drug-containing NPs toward specific cells. The use of peripherally conjugated targeting moieties for enhanced delivery of the NP systems is referred to as active targeting. This method was used to achieve high selectivity to specific tissues and to improve NP uptake into cancer cells and angiogenic microcapillaries. These compounds include an anticancer agent, a targeting moiety-penetration enhancer, such as receptors, receptor ligands, enzymes, antibodies, and surface modifications in active targeting methods. Another important feature of nanopharmaceuticals is the “triggered response”, which means that they can only begin to act in response to a specific activating signal (such as the influence of a magnetic field), allowing the NPs to release the drug locally once they have reached their target within the human body.

## 3. Brain Drug Delivery

The BBB represents one of the well-defined barriers separating blood from the neuronal parenchyma. The properties of this barrier are determined by the intercellular tight junction (TJ), which reduces the paracellular permeability and the passage of large molecules. The anatomical location of the BBB is into the cerebral capillary endothelium. The nomenclature “neurovascular unit” is now used to describe the combined activity and cohesion of microvessels together with the neurons and glia that surround them. Understanding the BBB is of fundamental importance and its implications is necessary for understanding pharmacodynamic of the therapies for neurological disease, since most large molecules that could promote a benefit in the treatment of brain diseases, from cancers to neurodegenerative diseases, do not cross the BBB or cross it in part or pass through small quantities before being degraded. New strategies are constantly being developed to overcome the BBB, especially through the bioengineered fusion of proteins that can be used as cotransporters or specific transporters to allow access to the brain parenchyma. However, the BBB is fundamental in supplying nutrients to the CNS, in allowing an outflow of waste molecules from the brain, in restricting the passage of ions and fluids through the blood and the brain, thus protecting the brain from significant fluctuations that may occur within the blood proper of ionic compounds of catabolites and metabolites. 

The endothelium of the cerebral capillaries is characterized by the presence of intercellular TJ as well as abundant cytoplasm, abundant mitochondria, and a low rate of endocytosis and pinocytosis. The structures of the interendothelial junction that allow the formation of the BBB are the TJs, and the others are a group of proteins with transmembrane domains, four, and with two extracellular loops defined, respectively, as occludins and claudins. Another important structure is the adherens junctions, which collaborate with the TJs and contain the vascular endothelial cadherin and the platelet endothelial cell adhesion molecule. Catenins represent the key point of interconnection between the intercellular structures and the cellular cytoskeleton. Other junctional elements include proteins of the immunoglobulin superfamily and are, respectively, the junctional adhesion molecules and the endothelial cell-selective adhesion molecule. The endothelial cytoplasm of the cerebral capillaries contains a large number of regulatory and signal proteins whose function is to modulate the interaction of membrane proteins with the active proteins of the cytoskeleton, such as zona occludens, calcium-dependent protein kinases. The anatomical site of the BBB is defined as the cerebral capillary endothelium, which also exhibits dynamic interactions with numerous other cell types. In fact, it is surrounded from pericytes and astrocyte stalks, which is often considered as the cells that connect the brain barrier to the cerebral environment. Therefore, a bidirectional interaction between the capillary endothelium of the CNS and its neighboring cells actually represents today the true definition of BBB (Figure 2). Here, there are important proteins that manage the maintenance of the structure both in a dynamic and physical sense, for example TGF-beta, the glial cell derived neurotrophic factor, and angiopoietin1. Therefore, since all these interconnections are present between all these cells that manage the passage, how the passage of the molecules through the blood–brain barrier really takes place depends on the size and biological properties of the molecules involved: the hydrophilic molecules can pass through the interendothelial spaces; lipophilic substances and gaseous particles, just like oxygen and carbon dioxide, instead directly cross the cellular endothelium. Specific transport proteins exist for different types of molecules, for example the glucose transporter GLUT1, the LAT1 transporter for amino acids, P-glycoprotein, and a whole other series of carriers. Some barrier transporters are also polarized, showing different properties inside and outside the barrier, thus allowing certain ionic passages. In fact, there is a genetic selectivity expressed in the cerebral capillaries: it allows the production of specific proteins. There is also a receptor-mediated transit mechanism to transport even larger proteins, such as plasma proteins including albumin, that would not otherwise pass.

The goal of absorption-mediated transcytosis is to deliver drugs via electrostatic interactions via NP systems functionalized with cell-penetrating peptides or cationic proteins. The adsorptive process, however, occurs in the blood vessels and other organs because it is a non-specific process. This makes it difficult to achieve therapeutic concentrations in the brain while also limiting drug distribution in non-target organs. CPPs and cationic proteins (e.g., albumin) are being studied to improve brain drug delivery via adsorptive-mediated transcytosis. CPPs have effectively delivered a wide range of cargo molecules/materials into cells, including small molecules, proteins, peptides, DNA fragments, liposomes, and NPs. TAT, a transcription factor involved in the replication cycle of the human immunodeficiency virus (HIV), has been shown to enter cells [22]. Transporters for nutrients for the brain are commonly overexpressed on the BBB and can be used for brain-targeted delivery [23]. Because the glutathione transporter is highly expressed on the BBB, researchers conjugated glutathione onto liposomes to deliver drugs to the brain. Systemic administration of glycosyl cholesterol derivative liposomes containing coumarin-6 resulted in a 3.3-fold higher Cmax with less cytotoxicity to brain capillary endothelial cells than conventional liposomes [24].

Because of its high specificity, receptor-mediated transcytosis (RMT) across the BBB has received more attention. Large molecules required for normal brain function are delivered to the brain via specific receptors expressed on BBB endothelial cells. After association between large molecules and theirs kindred receptors, they have shown transcytose transport via receptors [25]. Transferrin receptor (Tf-R) is a transmembrane glycoprotein that is overexpressed in GBM cells. Drugs can be targeted to the Tf-R using the endogenous ligand transferrin or antibodies directed against the Tf-R. Doxorubicin (DOX) loaded into Tf-R-NPs demonstrated anti-tumor activity, with a 70% longer median survival time than DOX solution-treated brain tumor-bearing rats [26]. Endogenous ligands could bind to receptors, reducing the binding efficiency of ligand-modified NPs. The strength of a drug’s effect is determined by the drug concentration at the receptor site, but the drug response may also be influenced by the number of receptors on a cell’s surface, second messengers (substances inside the cell), or regulatory factors that regulate gene translation and protein synthesis. Dimerization typically causes receptor activation. When a ligand binds to a receptor monomer, the extracellular domains of the receptor interact to dimerize the receptor, rendering it unavailable for a period of time [27]. The receptor binding theory serves as the foundation for the models used to describe the pharmacodynamic relationship. According to the classical receptor theory, the drug’s observed effects are the result of a series of biochemical and physiological changes that are triggered by reversible receptor binding. Once all of the receptors are occupied, the drug’s maximum effect is attained [28]. A drug’s interaction with a receptor is based on the arbitrary coupling of a ligand and a receptor, in accordance with the law of mass action. The “lock and key” model serves as an example of the fundamental idea. This will result in an effect that is proportional to the amount of ligand and receptor present. Since there may be a finite number of receptors and ligand concentrations may be higher than the affinity constant, the concentration–response relationship will behave in a way that is contrary to what would be predicted if the relationship were linear: higher concentrations will result in a smaller increase in response. A competitive ligand-binding model can typically explain the interaction between two and three ligands (drugs) at the same receptor-binding site. In this context, the term “competition” refers to the antagonistic interaction between two ligands that are capable of binding to the same receptor site. This is the reason why an endogenous natural ligand, such as the simple transferrin, could compete with a nanoparticle that had been conjugated with transferrin, lowering the amount of drug that could bind to a receptor and then be endocytized [27]. Antibodies against these receptors were developed to avoid this issue. Because the binding site of antibodies to receptors differed from that of ligands with receptors, ligand competition was avoided.

Ulbrich et al. created human serum albumin (HSA) NPs conjugated to transferrin or TR-mAbs (OX26) for loperamide delivery and demonstrated efficacy in transporting the drug to the brain in mice using OX26-conjugated HSA NPs. Because it binds to an extracellular domain of TR, OX26 mAb avoids competition with endogenous transferrin in the circulation system [29]. Aktas et al. recently designed OX26 mAb-bearing chitosan-PEG NPs and demonstrated that OX26 mAb is a critical functional moiety that allows NPs to cross the BBB [30]. LRP-1 and LRP-2 are ligand scavenger and signaling receptors with multiple functions. They can interact with a wide range of molecules and mediators, including ApoE, plasminogen activator inhibitor 1 (PAI-1), lactoferrin, heparin cofactor II, heat shock protein 96 (HSP-96), and engineered angiopeps [31].When associated with polysorbate 80-coated NPs, several drugs that do not cross the BBB, such as tubocurarine, loperamide, dalargin, 8-chloro-4-hydroxy-1-oxol, quinoline-5-oxide choline salt (MRZ 2/576), and DOX, show higher concentrations in the brain. When polysorbate 80, a nonionic surfactant, was conjugated on to NPs, it could adsorb ApoE in serum, and polysorbate 80-coated NPs have also been evaluated as a brain targeting delivery system by many groups [32,33]. Angiopeps are highly effective BBB targeting ligands, with angiopep 2 demonstrating increased transcytosis and parenchymal accumulation [34].

## 4. Carbon Nanomaterials

The family of carbon nanomaterials consists of different types of carbon-based structures. The family of carbon NPs includes many groups: fullerenes, carbon dots (CD), and carbon nanotubes (CNT), which in turn can be divided into single-walled (SWNT) and multi-walled (MWNT), graphene, and nanodiamonds (ND) (Figure 3). These different structures show different physical and electrochemical characteristics. Analytical applications have made extensive use of carbon-based nanoparticles. There are many different carbon-based materials that can be found and have been used in analytical processes. Any type of carbon nanomaterial has been used, but fullerenes and nanotubes have been the main focus of recent applications [35]. The fundamental building block in both instances is a layer of carbon atoms that are sp2-bonded together, with each atom being joined to three other carbon atoms in the bidimensional plane and by a weakly three-dimensional delocalized electron cloud. The ability to form charge-transfer complexes when in contact with electron donor groups and the good electrical conductivity is a result of this configuration, which is similar to that of graphene. Strong van der Waals forces, which significantly impede the solubility and dispersion of carbon-based nanoparticles, is also a result of this configuration [36]. According to previous reports, CNTs’ physicochemical properties can be tailored by doping them with foreign atoms, and their electronic, mechanical, and conductive properties can be significantly enhanced [37]. In addition to create carrying nanomolecules, carbon NPs could be selectively modified permitting electronic and thermal instability, when stimulated. Thermal properties can be useful in thermic therapy. One area of GB treatment research is hyperthermia therapy. With minimal damage to healthy tissues, hyperthermia is a type of treatment that exposes tissues to high temperatures in order to kill cancer cells. Targeted heating with nanoparticles can overcome the limitations of conventional treatments. The study focuses on a novel application of CNT that can selectively heat cancer cells by converting near-infrared light into heat. The shape, size, and volume fraction of the nanoparticles, as well as the thermal conductivities of the nanoparticles and the base fluid, are among the numerous factors that contribute to the fact that the theoretical description of the effective thermal conductivity of the nanoparticle-base fluid is still at an elementary level. A theoretical model has been put forth in relation to the estimation of the blood-CNT nanofluid’s effective thermal conductivity. Given the growing interest in targeting hyperthermia, there is still no widely accepted model for estimating the thermal conductivity of blood that contains CNTs in order to study the crucial issue of heat transfer during hyperthermic treatment. A theoretical model can offer helpful insight into the thermal conductivities of such bio-nanofluids because there have not been many experimental studies on the subject. The proposed hypothetical model considers the elements that make up blood, specifically blood cells and plasma. In a nutshell, the thermal conductivity appears to be improved by thinner, larger, and longer CNTs in combination with a significant amount of their concentration. To control the thermal conductivity of the blood-CNT nanofluid prior to or during the hyperthermic treatment of GBM or other types of cancer, these variables can be used selectively as design parameters [38]. Many studies in recent years on NPs are trying to identify which one of these carbon nanomaterials is more suitable for the transport of drugs conjugated to them or contained by them. Mendes et al. already in 2013 had noticed how drugs transported by carbon could find utility in the treatment of neurodegenerative diseases or brain tumors [39]. In 2017, Liu et al. began to create specific carbon-based nanostructures that target the brain. In fact, small carbon structures, if rationally functionalized on their surface, can cross the BBB and therefore transport drugs [40]. Recently, Porto et al. have shown that carbon nanomaterials have excellent thermal and electrical conductivity, strong adsorption capacity, high electrocatalytic effect, high biocompatibility, and high surface area [41]. These intrinsic characteristics would allow the structuring of pharmacological compounds and the simultaneous, potential, reduction in toxic effects. However, the ability to functionalize the surface of carbon nanoparticle structures must be well studied also on the basis of any direct and indirect toxicity that these nanoparticles can acquire.

Graphene is a sheet of carbon atoms arranged in a hexagonal grid. Each individual sheet is only one atom thick and therefore has a comparatively enormous lateral extent. For this reason, we consider the graphene as a two-dimensional material, in which there are only two dimensions of the plane, while the third is zero. Graphene has high mechanical strength properties, over 100 times more than steel because the atoms are linked together by very strong chemical bonds. Thanks to its particular chemical configuration, graphene possesses unique physical, electronic, optical, thermal, and mechanical properties. This molecule has shown promising applications not only in nanoelectronics, composite materials, energy technology, sensors, and catalysis, but also in biomedical research [42]. It has electrically conductive and thermally conductive properties superior to those of copper. It has a very high surface area to weight ratio. It is also totally waterproof, flexible, and can be made optically transparent and it is biodegradable. It can be differently modified in space into different two- and three-dimensional forms. The most common derived chemical forms are oxidized-type graphene, reduced-type graphene, nanoribbon graphene, and oxidized nanoribbon graphene, as well as quantum-dot graphene. Each of these show specific qualities in the transport of drugs.

CDs, on the other hand, are very tenacious carbon spheres, held together by covalent bonds of small dimensions, with dimensions less than 10 nm, studied since 2014 for the transport of drugs. Despite their small size, they are easy to craft. They are biocompatible, have a high capacity to penetrate and bind to receptors, are not very toxic, as demonstrated by the work of Shang et al., in which CDs were put in contact with stem cells [43].

CNTs are structures in which a tube made up of carbon hexagons is closed at the end by two hemifullerene caps. They have a high penetrating power and a large surface area. This means that many molecules can be conjugated to them, and all these properties can make them excellent candidates for the transport of anticancer drugs. SWNTs can be imagined as deriving from the process of rolling up a graphene plane on itself, closed at the ends by hemispherical caps of the fullerenic type. They have a high length/diameter ratio and for this reason they can be considered “almost” one-dimensional structures. MWNTs are nanotubes formed by multiple concentric SWNTs and are therefore called “multi-walled” nanotubes. The diameter of MWNTs is usually greater than that of SWNTs and increases with the number of walls. 

The NDs are of more recent discovery. The diamond proper is an allotropic form of carbon consisting of a crystalline lattice in which there are carbon atoms arranged with a tetrahedral symmetry. In this case, NDs are produced through controlled explosions inside closed chambers: the high pressure and temperature push the carbon atoms contained in the explosive substances to fuse together, thus obtaining tiny diamonds. They have a large surface area with a microscopic diameter between 2 and 8 nm. They are nanocrystals with a diamond-like structure, which gives them particular electronic and physical properties [44].

The fullerenes are spherical and resemble cages. Also known as buckminsterfullerene, it is a compound with a spheroidal polyhedral structure with 60 carbon atoms. Moreover, in this case their peculiar vesicle-like shape, formed by 12 pentagonal and 20 hexagonal faces with a total of 90 edges and 60 vertices, allows both surface conjugation and the possibility of internalizing molecules.

### Carbon Nanomaterials and Brain Tumors

Although there is still no selective drug for GB treatment, the attention of researchers has focused on this issue in recent years. Many studies have been carried out to evaluate in vitro and in vivo the possibility of using these NPs, even with non-classical anticancer drugs, but which, if linked to these carbon NPs, could become promising in the therapy against GB. Many anticancer drugs loaded into CNTs have been studied and their specificity against tumor cells or other tissues has been evaluated by conjugating them with specific target molecules. In reality, NTs by themselves can be absorbed through non-covalent hydrophobic interactions. However, the functionalization of carbon nanotubes with drugs or the addition of particular proteins that allow to define a membrane target allow the controlled release of drugs in the central nervous system [43]. Indeed, it has been seen that they can overcome the blood–brain barrier via a receptor-mediated endocytosis [45] (Table 1). 

To date, the mechanisms by which carbon materials can cross the BBB are still unclear. Receptor-mediated endocytosis is the best-known mechanism. CDs modified with transferrin and bound to epirubicin and temozolomide show a high capacity, compared to unmodified CDs, to bind to U87 glioma cells [46]. Another possible mechanism is passive diffusion. Thanks to their nanometric size, carbon nanometers cross the barrier by wedging themselves in the gap created between the endothelial cells and the BBB. The merger would favor the crossing of the barrier. Furthermore, the use of electric charges could further increase the ability to cross the BBB. Structuring a compound between CDs and cationic polyethyleneimine would increase BBB throughput [47]. A precursor of these studies was Zhao et al. in 2011 with the use of SWNTs conjugated to an immunostimulant oligonucleotide or cytosine-guanosine-motifs (CpG). This SWCNT-cPG was injected into mice with GL261-induced glioma observing an uptake within the tumor. However, this oligonucleotide is not currently considered an anticancer drug [48]. By contrast, in the following years many research groups have tried to combine classical and non-classical anticancer drugs with NTs (Table 2). Doxorubicin (DOX), which is not a first-line antiglioblastoma drug, has been successfully conjugated to be transported by MWCNT. In this study, the authors’ crafted molecule was formed via the oxidation of MWCNT, which was subsequently conjugated to Angiopeptin2 (Angiopep2) and polyethylene glycol (PEG). Once again, the success of the functionalization and transport of this system to glioma target cells was tested in vitro and subsequently in vivo, demonstrating once again how the created molecule MWCNT-PEG-Angiopep2 is more effective than single DOX [49]. Another similar result was obtained by another group of researchers with the use of oxaliplatin (OXA), conjugated to the BBB-penetrating peptide transcriptional activator (TAT), with biotin (B) and polyethyleneimine (PEI). This OXA-containing TAT-PEI-B copolymer was used in in vitro studies on murine glioma cells (C6) and human GBM cells (U87 and U251) to evaluate its absorption. In the subsequent in vivo study, the compound TAT-PEI-B-MCWTN@OXA proved to be much more cytotoxic than single OXA [50]. CNTs are considered as one of the most promising among carbon-based materials as drug carriers and the constant increase in studies represents their importance.

The attention to CDs is recent; in fact, they are little cited and represented in the literature in regard to their conjugation with anticancer drugs useful for GB treatment. Pioneering studies on the subject use the DOX. Transferrin-conjugated CDs bind DOX to form the molecule C-Dots–Trans–Dox, which has been shown in vitro to reduce the cell viability of different pediatric brain tumor cell lines [51]. DOX was also conjugated to polymer-coated carbon nanodots and IL6 fragments to give a specific target toward U87 glioma cells, which were later confirmed in vivo. It has been confirmed that this molecule crosses the BBB and selectively deeply penetrates GB cells allowing a gradual and constant release of drug. Furthermore, the presence of the IL6 fragment significantly reduces tumor cell growth, thus being able to conclude, thanks to the in vivo results, that this molecule increases the sensitivity toward DOX chemotherapy [52]. CDs have also been successfully conjugated with transferrin and epirubicin and TMZ for transport in GB cells, and as a result it has been noted that there is a synergistic effect of the triple-conjugated NP in reducing the viability of the tumor cell at a concentration lower than the same NP not conjugated with transferrin and compared to the two anticancer drugs used individually [46]. Even more recently CDs have been conjugated to gemcitabine with selective specificity for pediatric GBM cells. Moreover, in this case the molecule conjugated with transferrin make it possible to go beyond the BBB to reach the GBM cells. However, this preliminary study reported that a large amount of the drug is still needed to have an antitumor effect [53].

Graphene was accidentally discovered in 2004 by James and Novoselov [54]. Graphene compounds, like all other compounds of carbon NPs, can modify its properties with the different combinations of molecules. The family of graphene molecules includes a wide range of nanomaterials from oxidized graphene, reduced graphene, reduced oxidized graphene, graphene nanoribbons, oxidized graphene nanoribbons, ultrathin graphite, low-layer graphene, and so on. Among all the compounds under the study of nanomaterials, especially among carbon-based nanomaterials, graphene appears to be the most promising for biomedical applications thanks to its properties [55]. Already in 2012, Chen et al. had incorporated a chemotherapeutic agent, belonging to the nitrosourea family, into a molecule of oxidized graphene conjugated with polyacrylic acid (1,3-bis(2-chloroethyl)-1-nitrosourea) (BCNU). In vitro studies on GL261 glioma cells demonstrated the drug uptake via endocytosis and the greater efficacy of the drug conjugate compared to the virgin drug [56], but once again DOX is the drug most studied and used as an agent conjugated to carbon NPs. DOX molecules were created with pegylated oxidized graphene both with and without transferrin and studied on mouse models demonstrating how the DOX of the pegylated graphene oxide molecule associated with transferrin (PEG-GP-transferrin-Doxorubicin) reduced the tumor volume in rats [57]. Another molecule created with DOX is phospholipid-PEG-graphenenanoribbon-Doxorubicin, a pegylated graphene nanoribbon modified with phospholipids, studied in vitro against glioma U87 cells. This molecule once again demonstrated that the IC50 of DOX conjugated to a carbon-based NP was lower than unconjugated DOX [58]. In 2016 a study with Lucanthone, an off-the-shelf anticancer agent, allowed the creation of a molecule (Graphenenanoribbon-PEG-DSPE-Lucanthone) of oxidized graphene nanoribbon conjugated with 1,2-distearoyl-sn-glycero-3phosphoethanolamine-N-[amino(polyethylene glycol)] (PEG-DSPE) allowing a selective uptake from U251 glial cells without any effect on other neighboring cells in vitro [59]. More recently, DOX has always been associated with a graphene oxide molecule functionalized with lactoferrin, and the results of the in vitro uptake study on glioma C6 cells have documented that the major uptake of DOX was that of DOX conjugated and functionalized with lactoferrin (Lactoferrin-graphene oxide-iron oxide-Doxorubicin) [60]. Between 2021 and 2022, Szczepaniak’s group studied the direct effects of graphene molecules and therefore of graphene-conjugated NPs but not carrying anticancer drugs. These two studies demonstrated membrane potential changes and alteration in the viability of U87 tumor cells, thus continuing to hold promise for the utility of nanoparticle compounds in the treatment of GB [61,62]. Still, new studies are needed to select specific targets on GB cells and to choose promising new transporters to functionalize the NPs. Recent research has studied, in this regard, the use of curcumin conjugated to CDs [63]. 

The possibility of conjugating anticancer drugs to the surface of NDs and in particular DOX has been demonstrated since 2013. Although the greater efficacy of the conjugated drug has been established, in the literature there are still few studies with the use of anticancer drugs absorbed by ND in vivo [64]. The doxorubicin-polyglycerol-nanodiamond molecule in vivo and in vitro induces the autophagy of GB cells and also involves the expression of specific antigens, which cause an increase in the immunogenicity of GB cells. It could therefore be useful in reducing the immunosuppressive effect that occurs in patients affected by GB. Due to its high specific area, tunable surface structures, and biocompatibility, ND is a promising platform for biomedical applications such as imaging and drug/gene delivery. In order to specifically bind to the integrin receptor avb3, which is overexpressed in a variety of malignant tumor types, a drug carrier based on ND with a surface coating of PG was conjugated with cyclic ArgeGlyeAsp (RGD) peptide [65]. 

For what concerns fullerene, even if structurally similar to CNTs and similar in chemical and physical capacities, functionalized molecules associated with anti-tumor drugs have not yet been reported in the literature. A computer-based and computational-based predictive study on the use of fullerenes was conducted by Samantha and Das in 2017. This futuristic study demonstrates that anticancer drugs such as TMZ, procarbazine, carmustine, and lomustine can be absorbed non-covalently by the surface of the fullerene [66]. The conjugation of potent anticancer drugs with fullerene nanomolecules would be a great achievement especially in relation to the effects of single fullerene on nerve cells. It has, in fact, been demonstrated in vitro and in vivo in the last 14 years that fullerene is an excellent antioxidant and in general a neuroprotective drug [67,68].

## 5. Toxicity

Nanotoxicology studies the interactions of NPs with biological systems and the relationship between the physical and chemical properties of NPs with the induction of toxic responses. Factors such as particle size and shape, solubility and adsorption capability, exposure time, dose, aggregation and concentration, surface area and charge play a key role in the toxicity assessment of nanomaterials [69]. The shelf life, aggregation, leakage, and toxicity of materials used to make NPs are limitations for their use [70]. NPs are durable and can persist in the body for weeks, months, or even years, making them potentially toxic and limiting their use for repeated treatments. The increased production of reactive oxygen species (ROS) constitutes the basis of the toxicity of nanomaterials. ROS consist of reactive substances that derive from the partial reduction in oxygen, composed by uncoupled electrons localized in the outer shell of separate orbits [71]. This condition makes oxygen instable, creating free radicals through a partial reduction process. The increase in the production of ROS induced by exposure to NPs disrupts the antioxidant system, leading to oxidation of biological molecules, including proteins, lipids, and nucleic acids. Various mechanisms such as photocatalysis or chemical reactions of the ions released from NPs can trigger the direct production of ROS on the surface of the NPs themselves by inducing oxidative stress [72]. On the other hand, the presence of ROS can increase by an indirect mechanism such as a mitochondrial alteration linked to the exposure of NPs [72]. ROS show a greater reactivity being able to interfere with the physiological cellular functions [73]. Different kinds of ROS including superoxide anion (O^2−^), hydroxyl radical (OH), hydroperoxyl radical (HOˉ), hydrogen peroxide (H_2_O_2_), singlet oxygen (1O_2_), lipid peroxides (ROOH), and hypochlorous acid (HOCl) result in different toxic effects. Higher ROS production and accumulation damage all major macromolecules, such as proteins, lipids, and nucleic acids, and result in deleterious modifications of those molecules [74]. 

### 5.1. Carbon Nanomaterials and Oxidative Stress

Oxidative stress can be regarded as a common indicator of toxicity, which causes tissue injury and alters the regulation of metabolic pathways. Oxidative stress is a result of an imbalance between generated oxidants and antioxidants present in our body, which leads to cell apoptosis or degenerative process (Figure 4).

Despite the widespread diffusion of CNs, data on their potential toxicity remain controversial. The chemical and physical properties of CNs may vary, and they can differ in terms of morphology, purity, and structure based on the synthesis, functionalization, and purification methods used to produce them. Various experimental studies have demonstrated that oxidative stress induced by ROS generation is one of the responsible mechanisms for the toxic effect of carbon nanomaterials [75]. Excessive generation of ROS engendered protein carbonylation and DNA modification on exposure to GO at concentrations of 1–100 mg/L, ceasing the development of zebrafish embryos [76]. Oxidative stress and enzyme activity became enhanced after the injection of GO along with an increase in the HSP70 level in Acheta domesticus, a native grasshopper in South-Western Asia [77]. CNTs can cause toxicity through oxidative stress, mitochondrial damage, inhibition of protein synthesis, and cell death [78]. In vitro studies demonstrate increased ROS production and decreased glutathione levels as a function of a high concentration of SWCNTs [79]. CNT-mediated ROS generation leads to activation of cellular signaling pathways such as nuclear factor kappa B (NF-κB), activator protein-1 (AP-1), mitogen-activated protein kinase (MAPK), and protein serine-threonine kinase (Akt), which contribute to the proinflammatory response, tumor progression, and lung fibrosis [80]. In addition, ROS generated by CNT’s exposure induces genotoxic responses such as DNA strand breakage, formation of micronuclei or γH2AX foci, and chromosomal aberrations, possibly leading to carcinogenesis and fibrogenesis [81]. An interesting study reported ROS-activated, graphene-induced apoptosis that occurred through the MAPK and TGF-β signaling pathways [82]. SWCNTs induced oxidative stress and cellular toxicity in human epidermal keratinocytes [83], and incubation with SWCNTs triggered ROS generation and cell death along with NF-κB and p38 activation [84]. CNT’s fiber shape has been proposed as a critical factor in CNT-mediated toxicity. Kang et al. found that elongated-shaped MWCNTs were more cytotoxic than spherical-shaped carbon NPs, and led to a considerable elevation in intracellular ROS levels [85]. In a similar study, SWCNTs with 1–3 µm fiber length caused higher induction of intracellular ROS production as compared to those with 0.4–0.8 µm and 5–30 µm fiber length [86]. The type and degree of surface modification or functionalization of CNTs also influence CNT-mediated toxicity. Jiang et al. demonstrated that surface functionalization of MWCNTs with polyethylene glycol (PEG) reduces cellular uptake, intracellular ROS generation, and activity of oxidative stress-responsive pathways, such as p38 MAPK and NF-κB [87]. Many researchers have reported that metallic contaminants, such as iron, nickel, cobalt, and molybdenum, introduced during the manufacturing processes, are majorly responsible for CNT toxicity. In the presence of ferrous iron, H2O2 can be decomposed into hydroxyl radicals, which are the most destructive ROS with a strong oxidizing capacity to attack biomolecules in cells in a pH-dependent manner [88]. It has been reported that non-purified iron-rich SWCNTs are more effective in generating hydroxyl radicals than purified SWCNTs, and cause depletion of antioxidant molecules and accumulation of lipid hydroperoxides [89]. 

### 5.2. Carbon Nanomaterials and Genotoxicity

Genotoxicity represents any DNA or chromosomal damage that involves chromosome breaks, gene mutations, and rearrangements [78]. CNs can cross the cell membrane reaching the nucleus by simple diffusion or through the nuclear pores. The genotoxic effects of these materials can manifest themselves with a direct or indirect action on the DNA. Direct genotoxicity occurs when CNs interact with mitotic apparatus, and the consequent damages are linked to the specific phase of the cell cycle [90]. Indirect genotoxicity occurs as a result of oxidative stress and inflammatory response. Various factors are responsible for CNs’ interaction with genetic material including their capability to penetrate the nuclear membrane, size, and high affinity of some CNs to the G-C region of DNA sequences [91]. Aggregates of CNs can also cause deformation of the cell nucleus by altering the mitotic processes [92]. The increased production of ROS or the reduced functionality of DNA function repair are the main indirect mechanisms by which CNs can induce genotoxicity. CNs can cause ROS in the cells that may, through free radical attack, generate indirect oxidative damage to DNA. Such damages of the DNA base can cause mutations through mispairing in replication, leading to carcinogenesis [93]. Various experimental studies have demonstrated only reduced genotoxic effects induced by fullerenes and their derivatives [94,95,96]. The genotoxicity is usually caused by photo-induced DNA damage by interacting with NADH and the consequent generation of ROS [96]. C60 fullerene might interact with PMS2, RFC3, and PCNA proteins involved in the DNA mismatch repair pathway [97]. For NDs, it has been proposed that the induction of oxidative stress that may accompany long-term exposure is responsible for their toxicity [98]. Dworak et al. observed that ND-mediated oxidative stress may contribute to DNA damage on lymphocytes, which is susceptible to prolonged treatment to NDs [99]. Graphene can directly interact with DNA and induce genotoxicity through ROS production and oxidative DNA damage.

### 5.3. Carbon Nanomaterials and Neurotoxicity

Oxidative stress is the key factor in the potential damage due to the contact of carbon nanomaterials with the CNS. The brain, as a consequence of increased oxygen utilization and rapid metabolism, is particularly susceptible to ROS-induced toxicity [100]. Carbon nanoparticles can reach the CNS through the systemic, olfactory, and trigeminal pathways. The mechanisms of interaction between carbon nanomaterials and the CNS still appear to be unclear. It is probable that the shape, size, and duration of exposure to these materials can determine alterations or modifications of the cellular elements of the CNS through mechanisms that are still poorly understood. Within the brain parenchyma they can induce cytotoxicity, altering the molecular pathways and triggering chronic brain inflammation, microglia activation, and white matter alterations with increased risk for neurodegenerative diseases and stroke [101]. Recently, an experimental study showed that exposure to GO induced severe accumulation in the head region, increased ROS generation, decreased contents in dopaminergic, glutamatergic neurons, and different neurotransmitters and damage to AFD neurons in Caenorhabditis elegans [102]. Due to the dimensional characteristics, it has been observed that SWCNTs can penetrate inside the nerve cells by endocytosis and pinocytosis. The consequence of this process is the release of chemical mediators capable of inducing inflammatory processes, apoptotic processes, and oxidative stress [103]. In experimental animals, the introduction of MWCNTs would induce the release of cytokines, the activation of glial cells, and the triggering of inflammatory processes [104]. An experimental study demonstrated that GO exposure caused acute toxicity to zebrafish embryos. GO exposure induced a decreased hatching rate, disturbed locomotive activity, and upregulation of mRNA levels of genes related to nervous system, which suggested the potential risk of GO for developmental neurotoxicity. The GO treatment induced high levels of oxidative stress in vitro and in vivo [105]. It has recently been hypothesized that chronic exposure to carbon nanomaterials can cause neuronal cell death and the development of degenerative brain diseases. Furthermore, it has also been reported that upregulation of the Snca gene causes abnormal excitation of the neuronal system [106].

## 6. Discussion

The prognosis of cerebral gliomas still remains very poor. The commonly and widely accepted therapeutic protocol is a surgical approach followed by radio- and/or chemotherapy. Surgical techniques have evolved considerably in recent years thanks to the introduction of new technologies such as intraoperative imaging with MRI, CT, or ultrasonography, electrophysiologic monitoring, the visualization of tumor tissue with systemically injected fluorescent dye (5-aminolevulinic acid [5-ALA]), and surgery under local anesthesia with neurolinguistic cortical language mapping. Radiotherapy and chemotherapy treatments are represented by the STUPP protocol: administration of TMZ followed by radiotherapy. Nonetheless, the prognosis, in patients affected by cerebral gliomas, remains poor, not exceeding 15–20 months of survival. Various genes capable of triggering neoplastic activation processes and coding for key proteins have also been identified. Modern therapeutic approaches to brain tumors now aim to specifically target these biomolecules (VEGF, EGF, DKK…), thus attempting to slow down or stop the pathway underlying this protein. This approach, although very interesting, has some limitations. Initially, it is necessary to identify the key protein or suitable key proteins and try to target them selectively so as not to have side effects on healthy tissue. Furthermore, the presence of the BBB limits the access of these pharmacological compounds leading to an increase in the doses to be administered and the prolongation of the treatment.

This review focused on carbon-derived nanoparticles because of the recent interest on this field. A selective search on PubMed with the query “(carbon OR graphene OR fullerene OR nanotube OR nanodiamond) [title] AND nanoparticle” resulted in 47,831 manuscripts (Figure 5). 

By the way, the bias of the word “carbon” should be eliminated even if CD will not be searched: with the new query “(graphene OR fullerene OR nanotube OR nanodiamond) AND nanoparticle AND brain” (Figure 6) and more strictly with the query “(graphene OR fullerene OR nanotube OR nanodiamond) AND nanoparticle AND (brain[title] OR glioma[title] OR glioblastoma)” (Figure 7).

Research on drug-carrying nanoparticles in the central nervous system is still ongoing, and no fully commercially viable outcome has yet been obtained. There is still no favorite nanoparticle for this aim and much research is still in progress even for carbon-based and for the others type [107]. Table 3 presents the fundamental aspect of the research on nanoparticles.

In this study, we have reported some interesting experimental studies using carbon nanomaterials as possible therapeutic agents or carriers in the treatment of brain gliomas. The research carried out is substantially interesting and potentially valid. They are, of course, preliminary studies on cell lines of cerebral gliomas, but with very promising results. However, these studies have some limitations such as the lack of human trials and the lack of information on the potential toxic effects in human use. Recent experimental research, characterized by studies both in vitro and in vivo models, has clearly highlighted how the members of the family of carbon nanomaterials are able to induce oxidative damage, inflammation, activate different cell signaling pathways that can cause altered cellular responses and genotoxicity. Their mechanisms of toxicity involve oxidative stress and damage to the membrane, which can cause genotoxicity. The genotoxic effects of CNs have been identified by direct interaction with DNA to enhance DNA mutations. To date, the mechanisms responsible for the toxic effects of carbon nanomaterials are still poorly understood, considerably limiting their use. The specific molecular data relating to the interactions between the various types of carbon nanomaterials and their targets are still lacking. Therefore, it is necessary to correctly identify the physicochemical peculiarities responsible for the toxic effects. Various studies show that characteristics of the CNTs such as diameter, length, type, and structure, solubilizing agents (PEG, SDS), modification of CNTs (covalent or noncovalent functionalization), their aggregation behavior, and metal impurities, could play a key role in biological responses against nanomaterials [123,124,125,126]. The need for further experimental studies is evident in order to better investigate the mechanisms underlying the toxic phenomena of the CNs. In a recent, experimental study, the authors evaluated unfunctionalized graphene and carboxylated graphene for the potential of inducing oxidative stress. These data were validated in Daphnia magna by detecting the levels of oxidative stress biomarkers. The obtained results could, in theory, provide new indications on the induction mechanisms of the toxicity of graphene nanomaterials [124]. It appears essential to demonstrate, using healthy and transformed cell lines, the events of ROS formation, biocompatibility, and genotoxicity. When CNs are tested, it is necessary to characterize them in detail for the reliability, reproducibility, and comparability of data acquired in toxicological studies. In terms of toxicity models, comprehensive experimental information is required to be provided, including the target cell types, dispersion methods, exposure dosage, administration route in vivo [125]. The choice of materials is very important: materials with better physicochemical properties seem to have less toxic effects. Therefore, it appears appropriate in the preparation of the CNs to evaluate the presence of metallic impurities, to apply surfactant coating, and to check the length of the nanotubes [126].

## 7. Conclusions

Our study does not, of course, arrive at definitive results. Carbon nanomaterials represent, precisely because of their peculiarities such as the ease of passing cell membranes, the thermal properties, the large surface areas, and the easy modification with molecules, highly innovative materials and potentially suitable ones for being used in new therapeutic protocols against cerebral gliomas. On the other hand, there are also limitations to their use in humans, mainly linked to the onset of toxic phenomena affecting the nerve cells and the onset of inflammatory/oxidative processes. The new studies must now be directed to the search for new and more functional target molecules and, at the same time, through appropriate engineering, to the structuring of nanomaterials more suitable for human use.

## Figures and Tables

**Figure 1 cancers-15-02575-f001:**
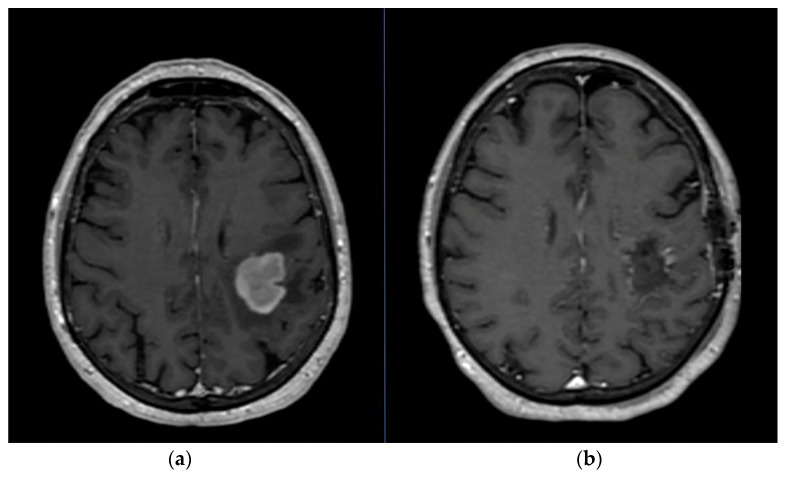
(**a**) Pre-operative MRI of a left frontal GBM; (**b**) Post-operative MRI. It is important to note that even if the resection is a supratotal resection, GBM has already infiltrated microscopically the nearest parenchyma.

**Figure 2 cancers-15-02575-f002:**
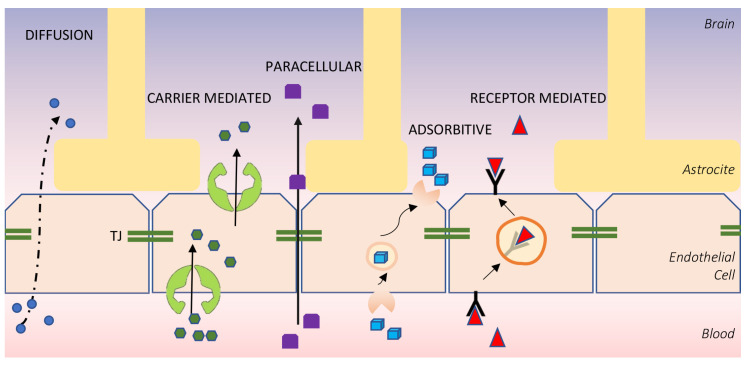
Scheme of BBB with its different transport mechanism. **Diffusion** (or **Transcellular Transport**): Some molecules cross the BBB forming transient “tunnels” in the endothelial cells, creating a passage toward the brain, through mechanisms that are not yet well defined. **Carrier-mediated transport**: It is selective for the substrate to be transported and based on steric interactions between the transporter and the transported molecule. The translocation across the membrane is linked to a conformational change of the carrier proteins because they are transmembrane proteins that cross entirely the plasma membrane, and more specifically to the opening/closing of a “channel” within the polypeptide. **Parecellular Transport**: It occurs through the hinges of the cell wall, i.e., the tight junctions that connect the endothelial cells of the BEE. **Adsorptive Transport** (or **Pinocytosis**): It is non-specific, i.e., the cell introduces small drops of extracellular matrix in an undifferentiated manner. This is possible because the material in question is dissolved in an aqueous solution. **Receptor-mediated Endocytosis**: It is regulated and specific. In this type of endocytosis, the cell recognizes its substrate.

**Figure 3 cancers-15-02575-f003:**
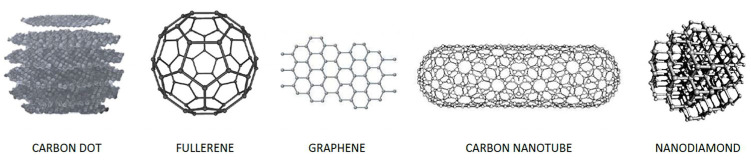
A schematic overview of the different types of carbon nanomaterials. **Carbon Dot** usually contain amorphous or nanocrystalline nuclei with predominantly sp2 hybridized carbon. The usual size of these objects is below 20 nm, and most often they are semiconductors. **Fullerenes**, represent a carbon allotropic structure that consists of an even number of sp2 hybridized carbon atoms. These are connected in 12 pentagonal and m hexagonal rings where m = (n − 20)/2, and where n is the total number of carbon atoms in the molecule. **Graphene** is a monolayer of sp2-hybridized carbon atoms with a hexagonal lattice (one layer of graphite). Each carbon atom has three σ-bonds and a p-bond outside the plane that can bind to adjacent atoms. **Carbon Nanotube** in its single wall nanotube has a diameter between 0.4 nm minimum and 6 nm maximum. The very high ratio between length and diameter (in the order of 10^4^) allows them to be considered as virtually one-dimensional nanostructures and confers peculiar properties on these molecules. As shown in the figure, their structure results from the union of the graphene and fullerene structures properly shaped. **Nanodiamond** is an allotrope formed by a network of sp3 hybridized carbon atoms arranged in a cubic area-centered lattice. Each carbon atom is bonded to the surrounding four atoms in the form of a tetrahedron. The diamond exists in two modifications, the hexagonal and the cubic type.

**Figure 4 cancers-15-02575-f004:**
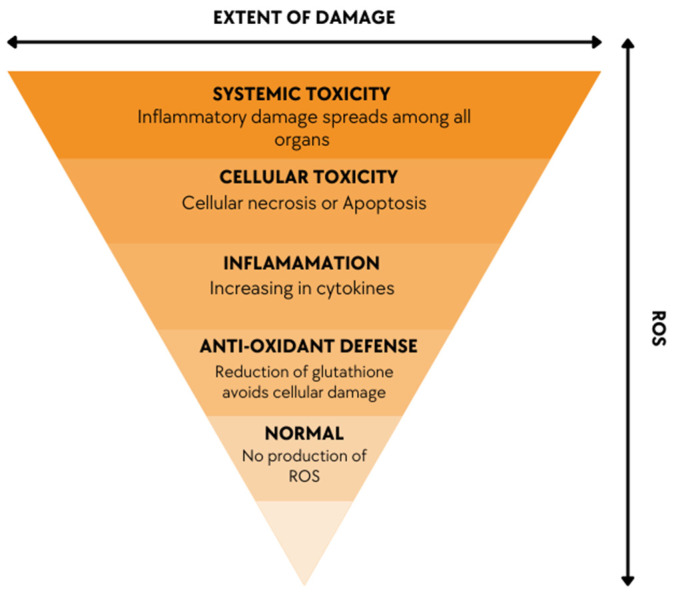
A diagram showing the direct proportionality between cellular and systemic damage and the fluctuation in Ros levels.

**Figure 5 cancers-15-02575-f005:**
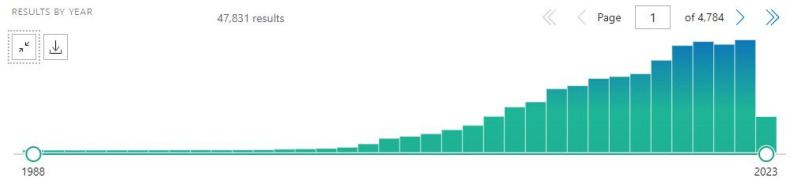
A representative diagram from PubMed, with the query “(carbon OR graphene OR fullerene OR nanotube OR nanodiamond) [title] AND nanoparticle”, showing the increasing interest in scientific research and publication in recent years.

**Figure 6 cancers-15-02575-f006:**
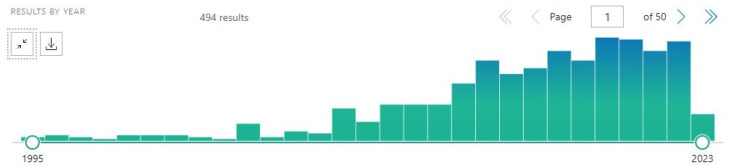
A representative diagram from PubMed, with the query “(graphene OR fullerene OR nanotube OR nanodiamond) AND nanoparticle AND brain”.

**Figure 7 cancers-15-02575-f007:**
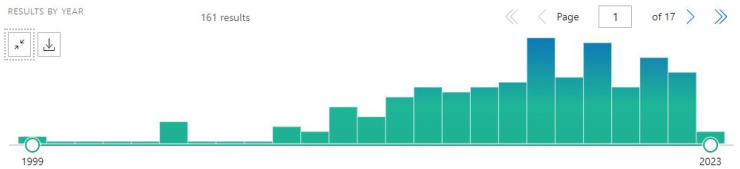
A representative diagram from PubMed, with the query “(graphene OR fullerene OR nanotube OR nanodiamond) AND nanoparticle AND (brain[title] OR glioma[title] OR glioblastoma)”.

**Table 1 cancers-15-02575-t001:** Resume of various compounds used for carbon nanoparticles functionalization.

Molecule	Cellular Ligand	Effect
Angiopeptin2 (Angiopep2)	Low-density lipoprotein receptor-related protein (LRP)	Transcytosis of BBB
Polyethylene Glycol (PEG)/ Phospholipid-PEG (PL-PEG)/ 1,2-distearoyl-sn-glycero-3phosphoethanolamine-N-[amino(polyethylene glycol)] (PEG-DSPE)		Avoid recognition by the reticuloendothelial system. Increases solubility and stability
Transcriptional Activator (TAT)		Inhibits Occludin Expression; reduce Occludin via matrix metalloproteinase-9
Biotin (B)	Sodium-dependent multivitamin transporters (SMVT)	Increase uptake inside cancer cell. (SMVT is overexpressed in cancer cell)
Polyethyleneimine (PEI)		Promote endosomal escape
Transferrin	Transferrin receptor	Transcytosis of BBB; endocytosis in GB (receptor is over expressed in brain tumor cells)
IL6 fragment	IL-6 receptor (IL-6R)	Transcytosis of BBB; endocytosis on GB; Block the IL-6-mediated pathway
Polyacrylic acid (PAA)		Enhances drug solubility; decreases drug hydrolysis rate
1,2-distearoyl-sn-glycero-3phosphoethanolamine-N-[amino(polyethylene glycol)] (PEG-DSPE)		Increases solubility and stability
Lactoferrin	Lactoferrin receptor	Transcytosis of BBB; endocytosis in GB (receptor is over expressed in brain tumor cells)
Polyglycerol		Reduce uptake and toxicity in macrophages; increases solubility and stability

**Table 2 cancers-15-02575-t002:** Resume of various NP-drug constructs cited in this article.

Molecule	Anti-Cancer Drug Delivered	Carbon NP	Author
SWCNT-cPG	*Cytosine-guanosine-motif* *	SWCNT	Zhao et al., 2011 [48]
MWCNT-PEG-Angiopep2	DOX	MWCNT	Ren et al. [49]
TAT-PEI-B-MCWTN@OXA	OXA	MWCNT	You et al. [50]
C-Dots–Trans–Dox	DOX	CD	Li et al. [51]
pCDPID	DOX	CD	Wang et al. [52]
C-dots-trans-temo-epi (C-DT)	Epirubicin + TMZ	CD	Hettiarachchi et al. [46]
CN–GM–Tf	Gemcitabine	CD	Liyanage et al. [53]
PAA–GO–BCNU	BCNU	Ox-Graphene	Chen et al. [56]
PEG-GP-transferrin-Doxorubicin	DOX	Ox-Graphene	Liu et al. [57]
phospholipid-PEG-graphenenanoribbon-Doxorubicin	DOX	Graphene nanoribbon	Lu et al. [58]
Graphenenanoribbon-PEG-DSPE-Lucanthone	*Lucanthone* *	Ox-Graphene nanoribbon	Chowdhury et al. [59]
Lactoferrin-graphene oxide-iron oxide-Doxorubicin	DOX	Ox-Graphene	Song et al. [60]
Doxorubicin-polyglycerol-nanodiamond	DOX	ND	Li et al. [65]
*Ideal molecule, still not synthetized*	*TMZ*; *Procarbazine; Carmustine; Lomustine*	Fullerene	*Samantha and Das* [66]

* Not considered anticancer drug or not in commerce.

**Table 3 cancers-15-02575-t003:** Resume of NP-drug constructs properties and their differences.

*Quality*	*Other Nanoparticles*	*Carbon-Based Nanoparticles*
*Immunotoxicity*	*Deficiency in immune system’s capacity* [108]	*High biocompatibility* [109]
*Acute toxicity*	*acute host damage* [108]	*Despite their accumulation in several organs and despite long half-life no acute toxicity is present* [110]
*Half-life*	*Quick Degradation* [111]	*Long half-life* [112,113]
*Tissue selectivity*		*Privileged interactions with neuronal cells (CNT)* [110]
*Interaction with neuron*		*Sustain neuronal survivor* [114]
*Bioavaibility*	*reduced bioavailability of the drug (Liposomes for example)* [115]	*able to cross the Blood–Brain Barrier (BBB) and accumulate inside the brain tissue* [109]
*Volume*		*High surface-to-volume ratio* [116]
*Shape*		*Ad hoc engineered* [110]
*Size*	*Include all particle 1–100 nm*	*Smallest dimension* [117]
*Thermal property*		*Thermal properties* [118,119]
*Chemical property*		*Chemical stability* [120] *Excellent electrical conductors*
*Mechanical property*		*Mechanical strength* [121]
*Production cost*		*Expensive production* [122]

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
