# Peer review of "Potential Role of Carbon Nanomaterials in the Treatment of Malignant Brain Gliomas"

_cancers, 2023, doi:10.3390/cancers15092575_

Round 1

Reviewer 1 Report

Currently nanoparticles have potentials for use in many areas including medicine. Understanding the mechanisms and issues related to nanoparticles is important for future works. This makes this review valuable. 

There are several areas on the paper that seems to be more of explanations from the author than the summary of the already published papers. 

Additionally, there are several English/grammar mistakes that makes it hard to follow what author trying to say.

Author Response

Reviewer #1: Currently nanoparticles have potentials for use in many areas including medicine. Understanding the mechanisms and issues related to nanoparticles is important for future works. This makes this review valuable. There are several areas on the paper that seems to be more of explanations from the author than the summary of the already published papers. 

Response: We appreciate the reviewer’s comment. Aim of this study was to report the most important published data regarding the potential use of carbon nanomaterials in the treatment of glioma. We are interested on this fields for some years; anyway, as reviewer suggested we modified the manuscript, inserting new and others references.    

Additionally, there are several English/grammar mistakes that makes it hard to follow what author trying to say.

Response: English has been extensively revised.

Reviewer 2 Report

The article mainly reviews the potential effectiveness of the use of carbon nanomaterials in the treatment of malignant gliomas and discusses the current progress of in vitro and in vivo researches of carbon nanomaterials-based drug delivery to brain. However, the manuscript has not been well prepared, I would like to suggest a major revision. I have several questions as follows.

(1)The properties and advantages of carbon nanomaterials are not well prominent. Especially, the thermal properties and electronic properties of carbon nanomaterials in the application of drug delivery and treatment of malignant gliomas are not well elaborated. Please provide.

(2)The organization of the manuscript should be adjusted. Specifically, Page 6, Line 211-240  should be put into “Nanotechnology” part. Besides, the content needs to be summarized more briefly. 

(3)For 5. Carbon Nanomaterials, which only lists the characteristics of various carbon nanomaterials and is not well related to drug delivery. It is suggested to discuss the application of carbon nanomaterials such as fullerenes and nano-diamonds in drug delivery.

(4)It is suggested to discuss the limitations of carbon nanomaterials in brain drug delivery, such as neurotoxicity and inflammatory stress. And possible solutions and future development need to be clarified briefly.

(5)The figures and charts of related studies cited by the article are not enough. Please add.

Author Response

Reviewer #2: The article mainly reviews the potential effectiveness of the use of carbon nanomaterials in the treatment of malignant gliomas and discusses the current progress of in vitro and in vivo researches of carbon nanomaterials-based drug delivery to brain. However, the manuscript has not been well prepared, I would like to suggest a major revision. I have several questions as follows.

Response: We thank reviewer for his suggestion. Accordingly, we included in the revised version all reviewer’s suggestions.

  • The properties and advantages of carbon nanomaterials are not well prominent. Especially, the thermal properties and electronic properties of carbon nanomaterials in the application of drug delivery and treatment of malignant gliomas are not well elaborated. Please provide.

Response: Description of Carbon Nanomaterials has been treated and inserted in section 4 (“Carbon-Nanomaterials”) pag. 10 line 318-353

  • The organization of the manuscript should be adjusted. Specifically, Page 6, Line 211-240 should be put into “Nanotechnology” part. Besides, the content needs to be summarized more briefly. 

Response: The section has been moved. Please note that another reviewer asked for “Chapters “BBB” and “NP” can be included in “Brain Drug Delivery” and “Nanotechnology,” respectively”, so both subsections have been greatly modified.

  • For 5. Carbon Nanomaterials, which only lists the characteristics of various carbon nanomaterials and is not well related to drug delivery. It is suggested to discuss the application of carbon nanomaterials such as fullerenes and nano-diamonds in drug delivery.

Response: A new text has been added to ND description.

  • It is suggested to discuss the limitations of carbon nanomaterials in brain drug delivery, such as neurotoxicity and inflammatory stress. And possible solutions and future development need to be clarified briefly.

Response: We, accordingly with these suggestions, inserted and discussed in a new section the toxicity of carbon nanomaterials, Section 5 (“Toxicity). In the subsection 5.1 we added a new paragraph “carbon nanomaterials and oxidative stress” and in the subsection 5.2, carbon nanomaterials and genotoxicity, as well in the subsection 5.3 carbon nanomaterials and neurotoxicity. Potential solution and future development have been added in the section 6, Discussion Section, pag 19, lines 729-760. 

  • The figures and charts of related studies cited by the article are not enough. Please add.

Response: Some figures and a new table as suggested have been accordingly added too.

Reviewer 3 Report

This comprehensive review focused on the potential application of carbon nanomaterials for brain drug delivery, especially for chemotherapy drugs. Overall, the manuscript is well conceptualized and organized. Moderate revision in the language may improve the readability of this article.   

Author Response

Reviewer #3: This comprehensive review focused on the potential application of carbon nanomaterials for brain drug delivery, especially for chemotherapy drugs. Overall, the manuscript is well conceptualized and organized. Moderate revision in the language may improve the readability of this article.   

Response: We really appreciate your comment. We are very satisfied of this hard work. A native language professor revised the manuscript.

Reviewer 4 Report

The authors focused on Carbon Nanoparticles, a drug delivery system for glioblastoma notorious for its poor prognosis. Brain drug delivery is a critical challenge in developing novel therapeutic modalities for malignant glioma. Readers will be interested in a review of the topic.

My comments on this manuscript are as follows.

1. Composition

The authors are recommended to describe introductive sections more concisely. 

Chapters “BBB” and “NP” can be included in “Brain Drug Delivery” and “Nanotechnology,” respectively.

2. References:

A few references are from the latest three years (2019~). Also, the publications on Carbon nanoparticles seem somewhat biased toward the publications of Chinese researchers.

3. The readers may want to know the strong points of carbon nanoparticles compared to other nanomaterials. It is easier to understand if the characteristics of Carbon NPs are shown in a Table compared to other NPs. 

4. The authors describe the safety of Carbon NP in the Discussion section. Safety is an important issue, so it should be written in a new chapter.

Author Response

Reviewer #4: The authors focused on Carbon Nanoparticles, a drug delivery system for glioblastoma notorious for its poor prognosis. Brain drug delivery is a critical challenge in developing novel therapeutic modalities for malignant glioma. Readers will be interested in a review of the topic. My comments on this manuscript are as follows.

            Response: We appreciate very much the interesting comment by the Reviewer and we modified the new version of manuscript as suggested.

  1. Composition

The authors are recommended to describe introductive sections more concisely. 

Chapters “BBB” and “NP” can be included in “Brain Drug Delivery” and “Nanotechnology,” respectively.

            Response: The “BBB” section has been moved to the section “Brain Drug Delivery”. For what concern “NP” there is no subsection with this name in this manuscript. Please note that another reviewer asked for: “(2)The organization of the manuscript should be adjusted. Specifically, Page 6, Line 211-240 should be put into “Nanotechnology” part. Besides, the content needs to be summarized more briefly”, so Brain Drug Delivery section has been greatly changed. 

  1. References:

A few references are from the latest three years (2019~). Also, the publications on Carbon nanoparticles seem somewhat biased toward the publications of Chinese researchers.

Response: We thank the reviewer ‘s suggestion. Our work of literature revision was very hard to settle. However, new and recent references have been added. We noted that a large amount of researches are made from Chinese authors.

  1. The readers may want to know the strong points of carbon nanoparticles compared to other nanomaterials. It is easier to understand if the characteristics of Carbon NPs are shown in a Table compared to other NPs. 

Response: We would thank the reviewer for the suggestion. It is difficult to resume all the characteristics of so many different families of nanoparticles. In every single family coexist different kind of nanoparticle that have different properties. A new table (Table 3) has been added, only to resume the strongest and most interest point in research on a specific nanoparticle family.

  1. The authors describe the safety of Carbon NP in the Discussion section. Safety is an important issue, so it should be written in a new chapter.

Response: We, accordingly with this suggestion, inserted and discussed in a new section the toxicity of carbon nanomaterials, Section 5 (“Toxicity). In the subsection 5.1 we added a new paragraph “carbon nanomaterials and oxidative stress” and in the subsection 5.2, carbon nanomaterials and genotoxicity, as well in the subsection 5.3 carbon nanomaterials and neurotoxicity.

Round 2

Reviewer 2 Report

The article mainly reviews the potential effectiveness of the use of carbon nanomaterials in the treatment of malignant gliomas and discusses the current progress of in vitro and in vivo researches of carbon nanomaterials-based drug delivery to brain. According to the responses, accurate correction and well-founded explanation had been finished. Additionally, the language had been polished carefully. Based on this , I would like to suggest accepting in present form.